# One-step direct oxidation of fullerene-fused alkoxy ethers to ketones for evaporable fullerene derivatives

Hao-Sheng Lin [1,2,6], Yue Ma [3,6], Rong Xiang[1], Sergei Manzhos[4], Il Jeon[1,5], Shigeo Maruyama [1] & Yutaka Matsuo [1,2,3 ✉]

Ketones are widely applied moieties in designing functional materials and are commonly obtained by oxidation of alcohols. However, when alcohols are protected/functionalized, the direct oxidation strategies are substantially curbed. Here we show a highly efficient copper bromide promoted one-step direct oxidation of benzylic ethers to ketones with the aid of a fullerene pendant. Mechanistic studies unveil that fullerene can serve as an electron pool proceeding the one-step oxidation of alkoxy group to ketone. In the absence of the fullerene pendant, the unreachable activation energy threshold hampers the direct oxidation of the alkoxy group. In the presence of the fullerene pendant, generated fullerene radical cation can activate the neighbour C–H bond of the alkoxy moiety, allowing a favourable energy barrier for initiating the direct oxidation. The produced fullerene-fused ketone possesses high thermal stability, affording the pin-hole free and amorphous electron-transport layer with a high electron-transport mobility.

[1] Department of Mechanical Engineering, School of Engineering, The University of Tokyo, Tokyo, Japan. [2] Department of Chemical System Engineering, Graduate School of Engineering, Nagoya University, Nagoya, Japan. [3] Hefei National Laboratory for Physical Sciences at the Microscale, School of Chemistry and Materials of Science, University of Science and Technology of China, Hefei, Anhui, China. [4] Centre Énergie Matériaux Télécommunications, Institut National de la Recherche Scientifique, Varennes, QC, Canada. [5] Department of Chemistry Education, Graduate School of Chemical Materials, Pusan National University, Busan, Republic of Korea. [6] These authors contributed equally: Hao-Sheng Lin, Yue Ma. ✉email: matsuo@photon.t.u-tokyo.ac.jp

Oxidation reactions, such as the direct oxidation of alcohols to aldehydes or ketones, are among the most critical and fundamental transformations in organic synthesis[1]. However, the oxidation methods are limited when alcohols are protected/functionalized with an alkyl group to form the alkoxy structure[2], which is mainly attributed to high activation energy barrier for directly converting the alkoxy group to ketone structure[3]. Consequently, the one-step direct oxidation of alkoxy to ketone has yet to be presented. Fullerene, a well-known intrinsically electron-deficient molecule, is prone to accept electrons affording the reduced fullerene anion species for versatile functionalizations[4,5]. With our interest in exploring the classical organic chemistry reaction under assistance of fullerene, and also inspired by the recent studies on fullerene radical cation ($C_{60}^{\bullet+}$)-mediated reaction[6,7], we conceived that $C_{60}^{\bullet+}$ should be feasible for the one-step oxidation of alkoxy group to ketone through the electron transfer activation.

Herein, we report that a copper (II) bromide promoted one-step direct oxidation of alkoxy to ketones with the aid of an oxidizable fullerene pendant (Fig. 1). Distinct from the unfavourable energy barrier in direct oxidation of alkoxy group to ketone, fullerene pendant serves as an electron pool for facilitating the electron transfer from the alkoxy structure to oxidant. Mechanistic studies indicate that the fullerene-assisted one-step oxidation involves two critical steps: (1) electron transfer from $C_{60}$ to Cu[II] affords $C_{60}^{\bullet+}$, and (2) the generated $C_{60}^{\bullet+}$ attracts electron density from the neighbouring C–H bond, contributing to the further electron transfer from the alkoxy structure to the fullerene cage. Meanwhile, the obtained fullerene-fused ketones are fabricated to the electron-transport layers through thermally deposition, which provides the photovoltaic devices with uniformly pin-hole-free electron-transport films. The reaction presented herein not only provides an understanding on one-step oxidation of alkoxy group to ketone, but also access the high-quality electron-transport layers through thermally evaporation.

## Results and discussion

**Reaction optimization.** Applied alkoxy substrate, indano[60] fullerene **1a**, was synthesized according to our previously reported fullerene-cation-mediated synthetic strategy[8]. The optimized reactions are summarized in Table 1, which includes the screening of the oxidants, reaction temperature and reaction time. The optimized conditions successfully achieved [60]fullerene-fused ketone **2a**, which was obtained in an isolated yield of 94%, with 4.0 equiv. $CuBr_2$ as oxidant in an ortho-dichlorobenzene (o-DCB) solution at 100 °C for 1.5 h under an argon atmosphere (entry 12). It is worth noting that **2a** can be obtained in equally high yield when the reaction was carried out in ambient environment (entry 14). Thus, this reaction appears to have high

efficiency and ease of operation, which would be useful for industrial-scale synthesis of fullerene-fused ketones.

**Substrate scope of the one-step oxidation reaction.** The substrate scope was further explored with some representative compounds. As shown in Table 2, this one-step oxidation reaction proceeded smoothly to afford **2a–d** in excellent yields. The optimized condition produced ketone **2a** with an isolated yield of 94% (entry 1). Methyl-substituted ketone **2b** was obtained in a similarly high yield of 93% (entry 2). To investigate the electronic effect of substituents, electron-withdrawing 4-fluoro- and electron-donating 4-methoxy-functionalized indano[60]fullerene substrates were synthesized. The corresponding ketones **2c** and **2d** were successfully isolated in excellent yields of 92% and 94%, respectively (entries 3 and 4) (for NMR spectra see Supplementary Figs. 14–26).

**Isotope-labelling experiments for determining the oxygen source.** Efficient oxidation of alkoxy indano[60]fullerene **1a** proceeded even under the argon atmosphere, indicating that the oxygen source was not directly from the air. Then, the oxygen source for this one-step oxidation was determined by performing the reaction in the presence of $^{18}O$ isotope-labelled water ($H_2^{18}O$) within a sealed tube (Fig. 2). The control experiment was performed without the addition of $H_2^{18}O$ under the optimized conditions. Then, the molecular weight of the product **2a** was measured by high-resolution mass spectrum (HRMS), which showed a mass-to-charge ratio (m/z) of 824.0260 corresponding to non-$^{18}O$-labelled **2a** (Fig. 2a). When the reaction was carried out in the presence of $H_2^{18}O$, a peak at m/z 826.0342 was clearly observed by HRMS, indicating that the obtained ketone contained $^{18}O$ in its carbonyl group (Fig. 2b). Therefore, the oxygen source for this oxidation reaction is $H_2O$ rather than the methoxy group or $O_2$ from the air. Notably, although an excess of $H_2^{18}O$ was used, the mass peak of non-$^{18}O$-labelled ketone **2a** can still be seen in the HRMS spectrum of the $^{18}O$-labelled ketone **2a**-($^{18}O$) (see Supplementary Figs. 1, 27, 28 and Supplementary Tables 1, 2 for details).

**Kinetic studies.** Reaction kinetics of this one-step direct oxidation reaction were measured to further understand the reaction characterics (Fig. 3). All reactions were performed under the same conditions except varying the temperature. The concentration changes over time were monitored by high-performance liquid chromatography (HPLC) (see Supplementary Figs. 2–5 and Supplementary Tables 3–6 for details). The change in concentration of reactant **1a** and product **2a** over time clearly indicated that this oxidation reaction reached equilibrium faster and gave higher yield when the reaction temperature increased (Fig. 3a, b). No by-products were formed during the transformation of **1a**–**2a**, demonstrating that this oxidation route has high selectivity and efficiency. The consumption ratio of **1a** was plotted on a logarithmic scale to determine the reaction order. The natural logarithm of the **1a** consumption ratio exhibited a strong linear time dependence, suggesting that this oxidation reaction exhibit the first-order characteristics (Fig. 3c). The rate constant (k) dramatically increased from $6.4 \times 10^{-4}\,mol^{-1}\,L^{-1}\,s^{-1}$ to $7.8 \times 10^{-3}\,mol^{-1}\,L^{-1}\,s^{-1}$ when the reaction temperature was increased from 353 to 375 K (Table 3). Next, the activation energy $E_a$, activation enthalpy $\Delta H^{\ddagger}$, activation entropy $\Delta S^{\ddagger}$, and activation Gibbs free energy $\Delta G^{\ddagger}$ were obtained from Arrhenius plots (ln k vs. 1/T) and Eyring plots (ln(k/T) vs. 1/T) on the basis of following equations, respectively (Fig. 3d)[9,10]:

$$\ln k = -E_a/RT + \ln A \qquad (1)$$

**Fig. 1 Working concept.** One-step direct oxidation of alkoxy groups to ketones with the fullerene as redox pendant.

**Table 1 Optimization of reaction conditions[a].**

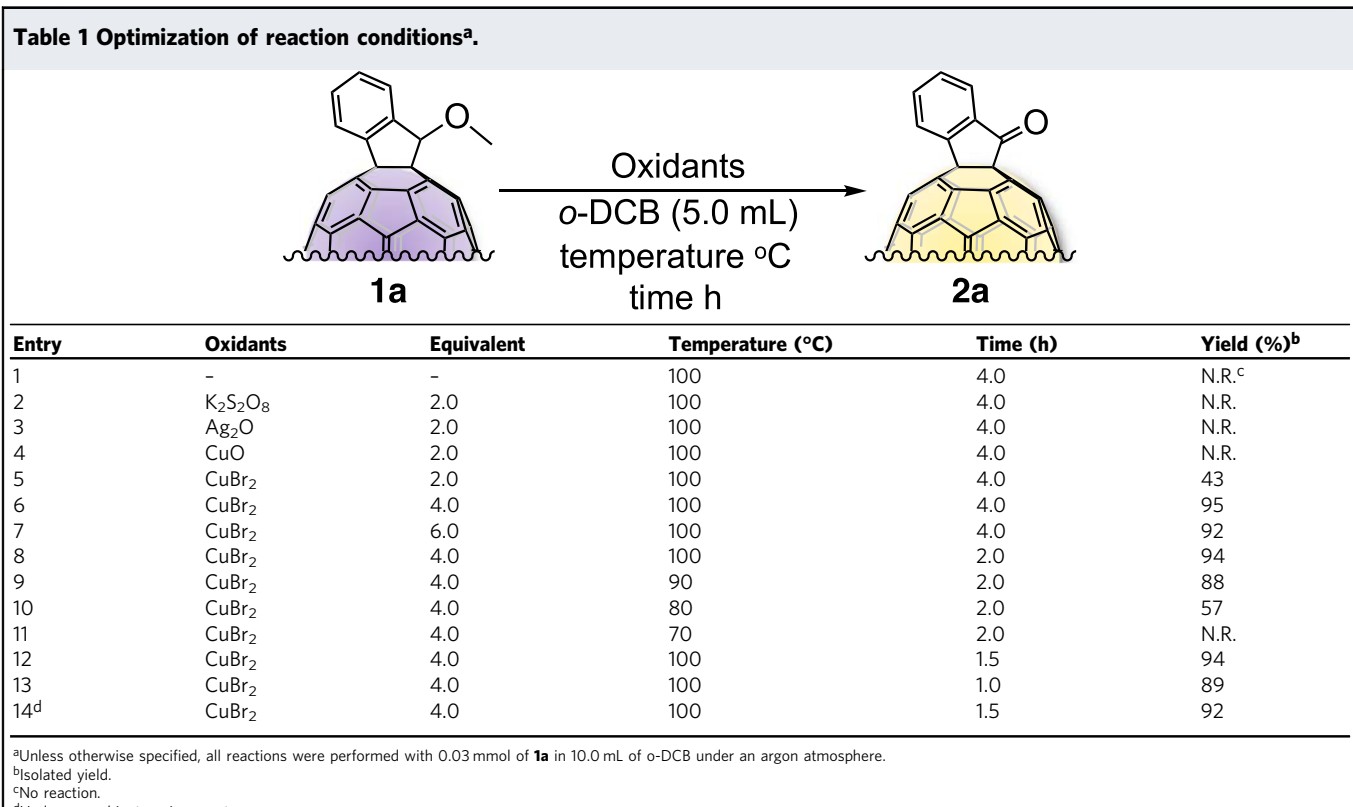

| Entry | Oxidants | Equivalent | Temperature (°C) | Time (h) | Yield (%)[b] |
|---|---|---|---|---|---|
| 1 | – | – | 100 | 4.0 | N.R.[c] |
| 2 | $K_2S_2O_8$ | 2.0 | 100 | 4.0 | N.R. |
| 3 | $Ag_2O$ | 2.0 | 100 | 4.0 | N.R. |
| 4 | CuO | 2.0 | 100 | 4.0 | N.R. |
| 5 | $CuBr_2$ | 2.0 | 100 | 4.0 | 43 |
| 6 | $CuBr_2$ | 4.0 | 100 | 4.0 | 95 |
| 7 | $CuBr_2$ | 6.0 | 100 | 4.0 | 92 |
| 8 | $CuBr_2$ | 4.0 | 100 | 2.0 | 94 |
| 9 | $CuBr_2$ | 4.0 | 90 | 2.0 | 88 |
| 10 | $CuBr_2$ | 4.0 | 80 | 2.0 | 57 |
| 11 | $CuBr_2$ | 4.0 | 70 | 2.0 | N.R. |
| 12 | $CuBr_2$ | 4.0 | 100 | 1.5 | 94 |
| 13 | $CuBr_2$ | 4.0 | 100 | 1.0 | 89 |
| 14[d] | $CuBr_2$ | 4.0 | 100 | 1.5 | 92 |

[a]Unless otherwise specified, all reactions were performed with 0.03 mmol of **1a** in 10.0 mL of o-DCB under an argon atmosphere.
[b]Isolated yield.
[c]No reaction.
[d]Under an ambient environment.

$$\ln(k/T) = -\Delta H^{\ddagger}/RT + \left[\ln(k_B/h) + \Delta S^{\ddagger}/R\right] \quad (2)$$

Here, $k$ is the rate constant, $T$ is the temperature, $R$ is the gas constant, $\ln A$ is a constant, $k_B$ is the Boltzmann constant and $h$ is the Planck constant. The results summarized in Table 1 indicate that this one-step direct oxidation has an $E_a$ of 120.6 kJ mol$^{-1}$, with an endothermic $\Delta H^{\ddagger}$ of 116.4 kJ mol$^{-1}$, a positive $\Delta S^{\ddagger}$ of 6.2 J mol$^{-1}$ K$^{-1}$.

**Mechanistic studies**. To gain more understandings on this one-step oxidation reaction, further investigations were carried out to understand the additional products and active intermediates. In situ proton nuclear magnetic resonance ($^1$H NMR) was applied to analyse in which form of the methyl group in **1a** exists the reaction (Fig. 4a). As shown in Fig. 4a, $^1$H NMR of **1a** clearly depicted a methyl peak with a chemical shift ($\delta$) at 4.252 ppm. After the reaction was fished, the in situ $^1$H NMR of reaction mixture indicated a disappearance of methyl peak in **1a**, while a new singlet peak appeared at $\delta = 2.619$ ppm. Compared to the methyl peak in methanol ($\delta = 2.827$), which we hypothesized as the potential leaved form of the methyl in this reaction, the reaction mixture showed substantially up-field shifted. In addition, the reaction mixture was found to be acidic, indicating the generation of acid during the reaction. Accordingly, we hypothesized that the $^1$H NMR signal ($\delta = 2.619$) of the reaction mixture should be derived from the methyl in CH$_3$Br[11], which was produced by the reaction between the generated HBr and MeOH leaving from **1a** especially reacting at high temperature. Meanwhile, when CH$_3$Br was formed, H$_2$O was simultaneously generated, which could then serve again as an oxygen source for this oxidation. Also, this result explained why non-$^{18}$O-labelled **2a** was still detected even when we used a large excess of H$_2{}^{18}$O.

Therefore, the methyl group in **1a** left in a methanol form, which further suggests that this oxidation reaction should involve a hemiketal intermediate. Moreover, the $^1$H NMR peaks depicted at the typical aromatic region indicates the obvious down-field shift of **2a** compared to **1a**, which is attributed to the electron deficiency of carbonyl group in **2a** (see Supplementary Fig. 6 for details). Also, the slightly positive $\Delta S^{\ddagger}$ of this reaction reasonably explained the increased disorder because of the additional products of MeOH and HBr. Besides the analysis of additional products that are generated during the reaction, further experiments were performed to confirm which active intermediate that mediated this one-step oxidation. The radical scavenger 2,2,6,6-tetramethyl-1-piperidinyloxy (TEMPO) was applied to confirm the generation of C$_{60}{}^{\bullet+}$ intermediate from the single-electron transfer between fullerene and CuBr$_2$ (Fig. 4b). When the reaction was run in the presence of 4.0 equiv. of TEMPO, the yield of **2a** was dramatically decreased from 95 to 8%. A further increase in amount of TEMPO to 10.0 equiv. stopped the reaction, suggesting that the electron transfer process was completely suppressed. Therefore, the one-step direct oxidation of the alkoxy is initiated by electron transfer from C$_{60}$ to CuBr$_2$, and C$_{60}{}^{\bullet+}$ plays a key role in the following oxidation steps.

Our mechanistic insights regarding the C$_{60}{}^{\bullet+}$ intermediate mediated one-step oxidation are provided in Fig. 5. Based on the above experimental results and our previous research[7], we considered that the oxidation of fullerene to C$_{60}{}^{\bullet+}$ through single-electron transfer in the presence of copper bromide demonstrate a critical role in this reaction. As depicted in Fig. 5, in this one-step oxidation, we hypothesized that the fullerene pendant in **1a** is initially oxidized by CuBr$_2$ via single-electron transfer, producing the key active specie, indano[60]fullerenyl radical cation **I**. Owing to the electron deficiency of C$_{60}{}^{\bullet+}$, the neighbouring C–H bond is then cleaved to generate neutral

**Table 2 Scope of reaction.**

| Entry | Products | Yield (%) | Entry | Products | Yield (%) |
|-------|----------|-----------|-------|----------|-----------|
| 1 | 2a | 94 | 3 | 2c | 92 |
| 2 | 2b | 93 | 4 | 2d | 94 |

Reaction conditions: all the reactions were performed with 0.02 mmol of **1a–d**, 0.08 mmol of CuBr$_2$ in 6.0 mL of o-DCB solution at 100 °C for 1.5 h under argon atmosphere.

radical **II** with the release of one proton, which then spontaneously reacts with the isolated bromide anion to form HBr. Next, CuBr$_2$ further oxidizes **II** to generate corresponding cation **III**, which undergoes nucleophilic addition by H$_2$O, producing hemiketal intermediate **IV**. Finally, [60]fullerene-fused ketone **2a** is produced through the loss of methanol and deprotonation. Meanwhile, the methanol produced can react with HBr to generate CH$_3$Br and H$_2$O, which then quickly reacts with benzyl cation **III** (see Supplementary Fig. 7 for details). Therefore, fullerene pendant can facilitate the one-step direct oxidation of the alkoxy group to ketone by serving as an electron pool.

**Computational studies**. To provide further support for the proposed mechanism, density functional theory (DFT) calculations were performed to understand the key species and reaction barriers (Fig. 6, see Supplementary Note 1 and Supplementary Table 7 for details). The DFT results indicated that the rate-determining step is proton transfer from H$_2$O to the methoxy group, which had computed potential energy barrier of 124.5 kJ mol$^{-1}$, in good agreement with the experiment value. It should be noted that Br$^-$ efficiently accelerated this proton transfer, as shown by DFT calculations in the absence of Br$^-$ (see Supplementary Fig. 8 and Supplementary Table 9 for details). In addition, the calculations showed facile oxidation of **1a** by copper (II) and relatively easy deprotonation of **I** to form the benzyl cation **III** and HBr, with an energy barrier on the order of 38.6 kJ mol$^{-1}$ (see Supplementary Fig. 9 for details). Therefore, CuBr$_2$ plays two roles in this one-step oxidation reaction: (a) oxidation of fullerene via electron transfer

with assistance of bromide anion, (b) proton transfer for formation of the hemiketal through the formation of Br$^-$.

**Performance of evaporable fullerene-fused ketone**. So far, fullerene derivatives have been demonstrated as versatile and high-performance n-type semiconductive materials in both organic solar cells and recently boosted perovskite solar cells[12–14], but the high-performance fullerene derivatives electron-transport materials have never been achieved using vacuum-deposition process[15–17]. Accordingly, both the indano[60]fullerene **1a** and the produced fullerene-fused ketone **2a** were further processed through the vacuum-deposition process to fabricate the electron-transport layer. HPLC analysis of vacuum-deposited **1a**-film indicated that **1a** is instable in the vacuum-deposition process, which affords a mixture of thermally decomposed compounds (Fig. 7a). Meanwhile, thermogravimetric analyses (TGA) showed that **1a** has an initial decomposing temperature at 258.9 °C with a 18.3% weight loss (see Supplementary Figs. 9 and 10 for details). Consistent with the HPLC analysis of vacuum-deposited **1a**-film, such dramatically high weight-loss ratio of **1a** can be attributed to a mixture of decomposed compounds. We observed that no thermally decomposed components were detected when vacuum-depositing **2a**, which indicated that the fullerene-fused ketone has a better stability (Fig. 7b). Further TGA manifested that **2a** show high thermal stability with an initial decomposing temperature at 409.5 °C, which is much thermally stable than that of **1a** or PC$_{61}$BM (see Supplementary Fig. 11 for details). Transmission electron microscopy (TEM) was carried to compare the morphology of spin-coated and vacuum-deposited **2a**-films,

respectively. Figure 7c indicated that the spin-coated **2a**-film show obvious pinholes with substantial crystalline found in the selected area electron diffraction (see Supplementary Fig. 12a for details). In stark contrast, the vacuum-deposited **2a**-film exhibits a highly uniform and amorphous morphology, which benefits the high electron-transport performance (Fig. 7d and See Supplementary Fig. 12b for details)[15,18]. To evaluate the charge carrier mobility of fullerene-fused ketone **2a**, space-charge-limited current (SCLC) measurements were applied to compare the trap-filling limit voltage ($V_{TFL}$) and trap density ($n_t$) of spin-coated and vacuum-deposited **2a**-films, respectively (Fig. 7e, f). In well accordance with TEM observations, the spin-coated **2a**-film showed more defects with higher $V_{TFL}$ (1.49 V) and $n_t$ ($1.4 \times 10^{18}$ cm$^{-3}$), compared with $V_{TFL}$ (1.01 V) and $n_t$ ($9.3 \times 10^{17}$ cm$^{-3}$) of the vacuum-deposited **2a**-film. Moreover, additional SCLC measurements further compared vacuum-deposited C$_{60}$- and **2a**-films (Fig. 7g, h). The **2a** film exhibited an equally high electron mobility ($2.16 \times 10^{-6}$ cm$^2$ V$^{-1}$ s$^{-1}$) compared with C$_{60}$ film ($2.33 \times 10^{-6}$ cm$^2$ V$^{-1}$ s$^{-1}$), which suggests that fullerene-fused

ketone can be applied as an efficient electron-transport layer to replace the pristine [60]fullerene in perovskite solar cells. Besides the electron-transport mobility comparison of **2a**, the lowest unoccupied molecular orbital (LUMO) energy level of **2a** is also evaluated through cyclic voltammetry (see Supplementary Tables 9 and 10 for details). Compared with LUMO levels of alkoxy substrates **1**, fullerene-fused ketones **2** show ~0.1 eV lower LUMO levels due to the electron-withdrawing property of carbonyl group. It should be noted that fullerene-fused ketones indicate the lowest LUMO energy levels among fullerene derivatives by far, which doubtlessly are compatible with common perovskite materials such as CH$_3$NH$_3$PbI$_3$. Meanwhile, DFT calculations demonstrate that fullerene-fused ketones have a deep enough highest occupied molecular orbital energy level, which is capable of blocking holes from recombining with electrons in photovoltaics (see Supplementary Table 11 for details). In addition, the UV–vis spectra were applied to compare the visible light absorption of vacuum-deposition fabricated **2a**-film and C$_{60}$ film (see Supplementary Fig. 13 for details). The absorption spectrum of **2a** showed an obvious blue-shift compared to that of C$_{60}$ at two typical fullerene absorption positions around 250 and 330 nm, which was further confirmed through DFT computation. This significant blue-shift is contributed by the strong electron-withdrawing carbonyl group in **2a**. More interestingly, compared to C$_{60}$, **2a** shows a much weak light absorption at a range of 400–650 nm, where is a typical absorption range for perovskite materials, providing a better light absorption for the perovskite materials when using **2a** as the electron-transporting layer in a normal-type structure.

In summary, here we report a facile CuBr$_2$-promoted one-step direct oxidation of alkoxy to ketone with the aid of an oxidizable fullerene pendant. The mechanistic investigation demonstrates in situ generated fullerenyl radical cation (C$_{60}$$^{\bullet+}$) behaves as an electron pool to facilitate the one-step direct oxidation: (a) initiating oxidation via electron transfer from C$_{60}$ to CuBr$_2$ to form C$_{60}$$^{\bullet+}$ and (b) activating cleavage of the neighbouring C–H bond by withdrawing electrons from the bond and subsequently affording the key hemiketal intermediate. Moreover, we found

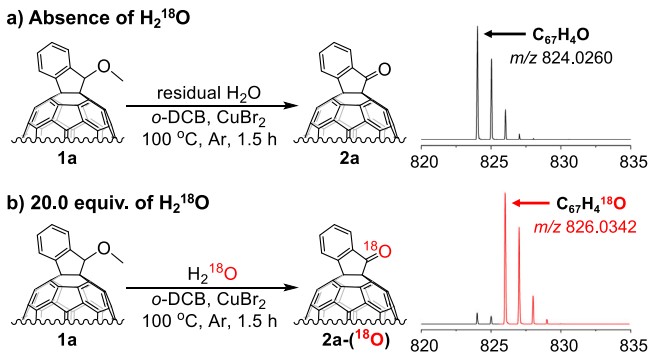

**Fig. 2 ¹⁸O isotope-labelling experiments.** Reaction conditions: **1a** (3.0 mg, 3.6 μmol), CuBr$_2$ (4.0 equiv.), o-DCB (3.0 mL) at 100 °C for 1.5 h in a sealed tube. **a** Absence of H$_2$$^{18}$O. **b** 20.0 equiv. H$_2$$^{18}$O.

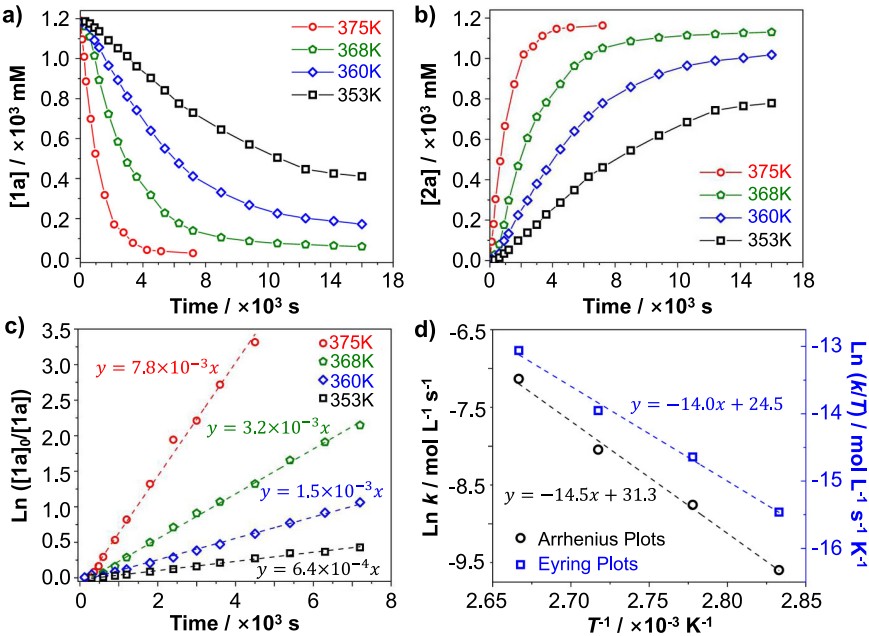

**Fig. 3 Reaction kinetics.** Reaction conditions: **1a** (3.0 mg, 3.6 μmol), CuBr$_2$ (4.0 equiv.), o-DCB (3.0 mL). **a** Concentration of **1a** over time at different temperatures. **b** Concentration of **2a** over time at different temperatures. **c** Plots of ln([**1a**]$_0$/[**1a**]) over time at different temperatures, where [**1a**]$_0$ and [**1a**] are the initial and remaining concentrations of **1a**, respectively. **d** Arrhenius plot (black) and Eyring plot (blue) for this one-step oxidation reaction.

**Table 3 Kinetic parameters of the one-step oxidation reaction[a].**

| Temp. (K) | Rate constant $(mol^{-1} L^{-1} s^{-1})$ | $E_a$ $(kJ\ mol^{-1})$ | $\Delta H^{\ddagger}$ $(kJ\ mol^{-1})$ | $\Delta S^{\ddagger}$ $(J\ mol^{-1}\ K^{-1})$ | $\Delta G^{\ddagger}$ $(kJ\ mol^{-1})$ |
|---|---|---|---|---|---|
| 353 | $6.4 \times 10^{-4}$ | 120.6 | 116.4 | 6.2 | 114.1 |
| 360 | $1.5 \times 10^{-3}$ | | | | |
| 368 | $3.2 \times 10^{-3}$ | | | | |
| 375 | $7.8 \times 10^{-3}$ | | | | |

[a]Activation free energy $\Delta G^{\ddagger}$ was estimated from $\Delta H^{\ddagger}$, $\Delta S^{\ddagger}$ and temperature at 375 K according to the equation $\Delta G^{\ddagger} = \Delta H^{\ddagger} - T\Delta S^{\ddagger}$[19].

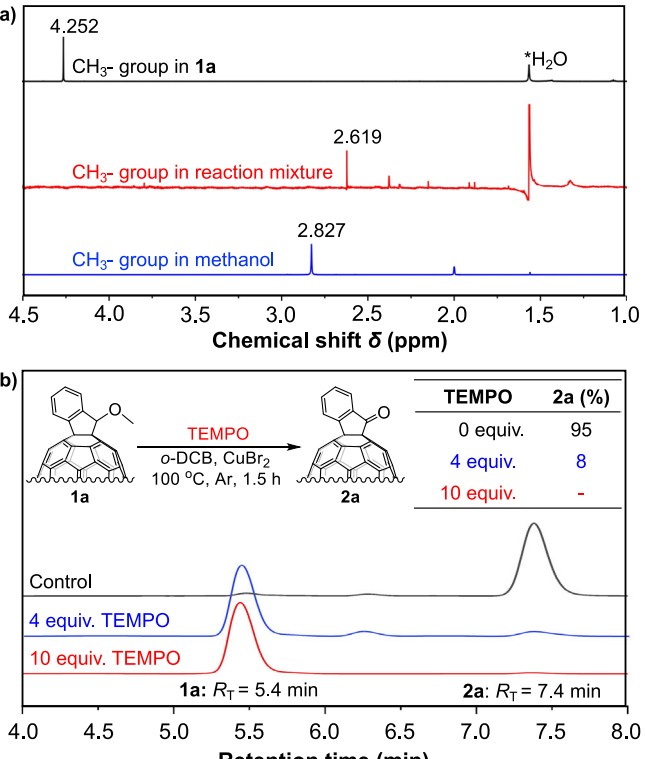

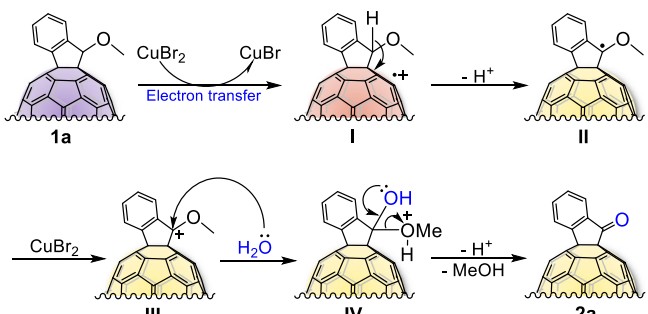

**Fig. 4 Mechanistic studies. a** $^1$H NMR with the $H_2O$ as internal reference located at up-field for identifying the methyl structure of **1a** (black), reaction mixture (red) and methanol (blue). **b** Experiments in the presence of different amount of TEMPO as radical scavenger.

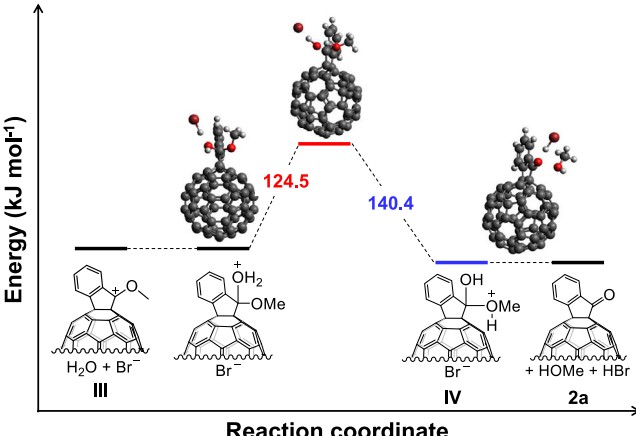

**Fig. 6 DFT calculations.** DFT calculation for the key step from intermediate **III** to **2a** in the presence of Br⁻.

of alkoxy to ketone and in fullerene cation chemistry, but also provide an evaporable fullerene material for high-performance electron-transport material in perovskite solar cells.

## Methods
**General procedures for the one-step oxidation reaction.** For fullerene-fused ketones **2**, all reactions were performed by using 0.03 mmol of **1**, 0.12 mmol anhydrous $CuBr_2$ (Sigma-Aldric) in 10.0 mL of anhydrous o-DCB under an argon atmosphere or open-air condition at 100 °C for 1.5 h. After reaction was over, resulting mixture was directly filtered through a silica gel plug to remove insoluble salt, and then evaporated in vacuo to remove the excess solvent. Finally, the residue was further separated on a silica gel column with $CS_2$ or $CS_2$/dichloromethane as eluents to afford products **2** (see Supplementary Methods for details).

**$^{18}O$ isotope-labelled experimental procedure.** 3.0 mg (3.6 μmol) of **1a** and 1.5 μL of $H_2^{18}O$ (0.072 mmol, 20.0 equiv) were added to 3.0 mL of anhydrous o-DCB solution in the presence of $CuBr_2$ (3.2 mg, 14.4 μmol, 4.0 equiv). After being vigorously stirred at 100 °C for 1.5 h with a tiny sealed tube, resulting mixture was directly filtered through a silica gel plug to remove insoluble materials. Finally, the filitrate was condensed in vacuo for the following HRMS measurement.

**TEMPO experimental procedure.** 3.0 mg (3.6 μmol) of **1a**, 3.2 mg of $CuBr_2$ (14.4 μmol, 4.0 equiv) and TEMPO (2.3 mg, 4.0 equiv; 5.6 mg, 10.0 equiv) were added to 3.0 mL of anhydrous o-DCB solution. After being vigorously stirred at 100 °C for 1.5 h under the argon atmosphere, resulting mixture was directly filtered through a silica gel plug to remove insoluble materials. Finally, ca. 50 μL of filitrate was directly loaded on HPLC to analyse results.

**Measurement of trap-filling limit voltage ($V_{TFL}$) and trap density ($n_t$).** $V_{TFL}$ and $n_t$ were evaluated based on SCLC using charge carrier only devices with a configuration of ITO/fullerenes (75 nm)/Au (60 nm). The $V_{TFL}$ and $n_t$ were calculated from the following equation: $V_{TFL} = \frac{n_t e d^2}{2 \varepsilon_0 \varepsilon_r}$, where $e$ is electric charge ($1.602 \times 10^{-16}$ V m⁻¹), $\varepsilon_0$ is the vacuum permittivity ($8.85 \times 10^{-19}$ V m⁻¹), $\varepsilon_r$ is the relative permittivity taken as 46.9 and $d$ is the thickness of the fullerene layer. The thickness of the fullerene layer was measured by using cross-sectional scanning electron microscopy. The experimental dark current density was measured under an applied voltage swept from 0 to −5 V.

**Measurement of electron mobility.** The electron-transport layer only device with a configuration of ITO/Fullerenes (30 nm)/Al (80 nm) was fabricated to evaluate the electron carrier mobility of synthesized fullerene-fused ketone. The mobility was determined by fitting the dark current to a model of a SCLC, which is described by the equation: $J_{SCLC} = \frac{9 \varepsilon_0 \varepsilon_r \mu V^2}{8 L^3}$, where $J_{SCLC}$ is the current density, $\mu$ is the electron mobility, $\varepsilon_0$ is the vacuum permittivity ($8.85 \times 10^{-19}$ V m⁻¹), $\varepsilon_0$ is the relative permittivity taken as 46.9, $L$ is the thickness of the fullerene layer and $V$ is the effective voltage. The thickness of the fullerene layer was measured by using cross-sectional SEM. The experimental dark current density was measured under an applied voltage swept from 0 to −3 V.

**Fig. 5 Proposed mechanism.** Plausible mechanism for $CuBr_2$-promoted one-step direct oxidation of alkoxy group to ketone.

that produced fullerene-fused ketone can form the high-quality electron-transport film using the vacuum-deposited process. Therefore, this reaction will not only provide a useful method in fundamental organic chemistry regarding the direct oxidation

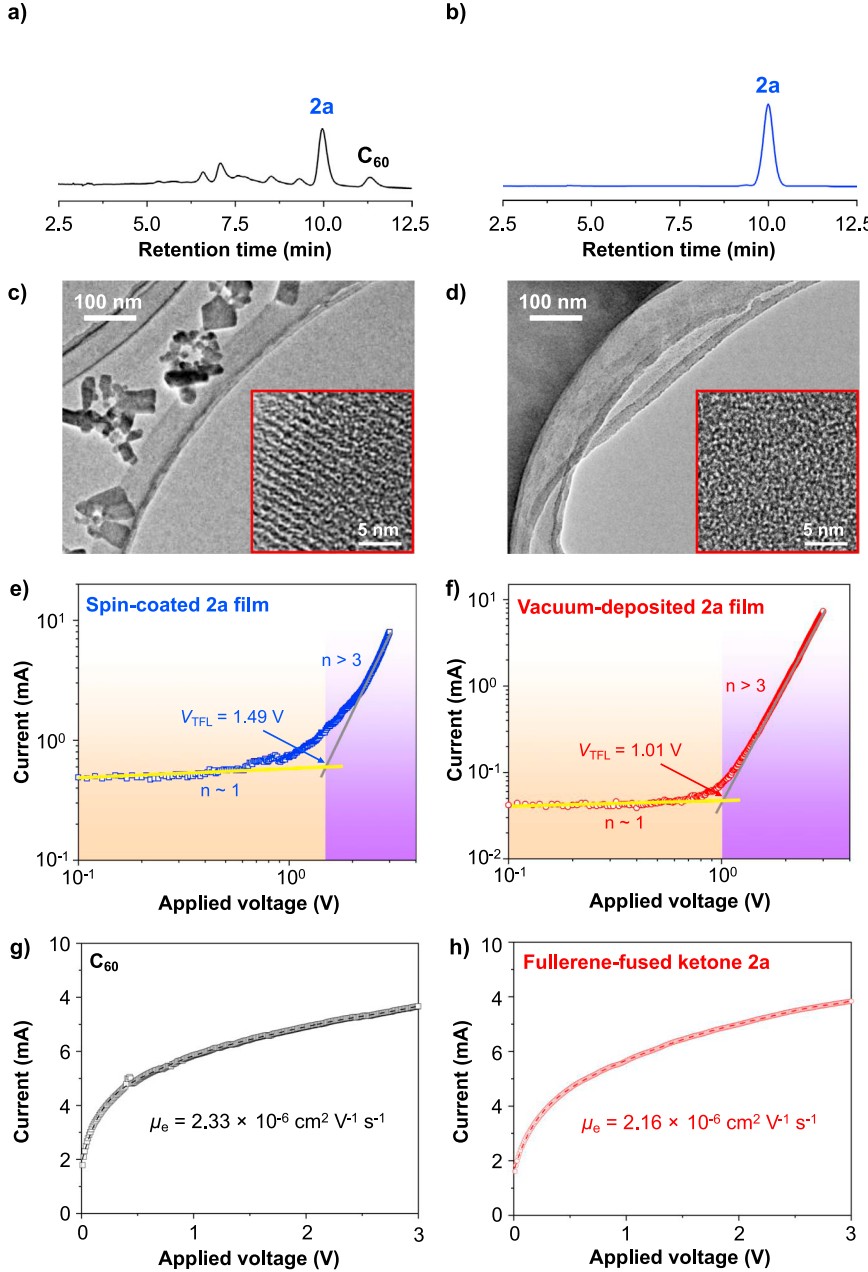

**Fig. 7 Properties of evaporable fullerene-fused ketone material.** HPLC analyses of **a** Vacuum-deposited indano[60]fullerene **1a**-film. **b** Vacuum-deposited [60]fullerene-fused ketone **2a**-film. TEM observations of **c** Spin-coated **2a**-film and **d** Vacuum-deposited **2a**-film with the magnification as an inset. Trap density measurement of **e** Spin-coated **2a**-film and **f** vacuum-deposited **2a**-film. The SCLC based electron mobility measurement of **g** vacuum-deposited $C_{60}$-film and **h** Vacuum-deposited **2a**-film.

## Data availability
The authors declare that the data supporting the findings of this study are available within the paper and its supplementary information files, or from the corresponding author upon reasonable request.

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

## Acknowledgements
We gratefully acknowledge funding from Japan Society for the Promotion of Science (JSPS) KAKENHI Grant Numbers JP18H05329, JP20H00220 and JP20J13124. Also, we would like to thank Strategic International Research Cooperative Program (SICORP, Grant Number JPMJSC18H1) and CREST (JPMJCR20B5), Japan Science and Technology Agency (JST).

## Author contributions
H.-S.L. and Y. Ma. are contributed equally to this work. H.-S.L. and Y. Mat. conceived the idea, analysed the data and wrote the manuscript. H.-S. L., Y. Ma. and I.J. conducted experiments, R.X. and S.M. conducted TEM observations. S.M. conducted DFT calculation.

## Competing interests
The authors declare no competing interests.
