## [Peer Review File · Communications Chemistry]

Reviewers' comments:

Reviewer #1 (Remarks to the Author):

In this manuscript, the authors reported the synthesis of fullerene-fused ketone via one-step reaction. The optimization of reaction conditions, scope of the reaction, and reaction mechanism are well studied and presented. This is a nice work and should be of interests to chemistry and organic electronics community. Therefore, I am happy to recommend the publication of this manuscript.

Minor revision suggestion: In Table 1, It is confused the readers of "t h" in the reaction condition.

Reviewer #2 (Remarks to the Author):

In this manuscript, Yutaka Matsuo et. al developed a highly efficient copper bromide promoted one-step direct oxidation of alkoxy to with the aid of a fullerene pendant. They further investigate the mechanism of the reaction, their results indicating that fullerene can serve as an electron poor proceeding the one-step oxidation of alkoxy group to ketone. They also prepared the thin films of the fullerene-fused ketone, affording the pin-hole free and amorphous electron-transport layer and measured the electron-transport mobility by SCLC method. Overall, the synthesis method is simple while efficient, the characterizaion is comprehensive. I suggest to be published in communications chemistry after these comments have be addressed.

- 1) The comparison of the proton NMR of the 1a and 2a should be added.
- 2) What is the temperature of the NMR of the 2a, 2b, 2c and 2d have been done should be added in the SI, at room temperature or high temperature?
- 3) The TGA curve of 1a is weird, can the authors explain what is the first step of the curve?
- 4) I suggest to add the absorption spectra, the CV data and the energy levels of the fullerene, 1a and 2a both in solution and thin film, and make a comparison and explanation if there are different.
- 5) The authors also made the spin-coated and vacuum-deposited thin films, what solvent has been used to get the spin-coated thin films?
- 6) Between line 242 and 246, the authors mentioned that "Fig. 7c indicated that the spin-coated 2a-film show obvious pinholes with substantial crystalline found in the selected area electron diffraction (See Supplementary Fig. 10a for details). In stark contrast, the vacuum-deposited 2a-film exhibits a highly uniform and amorphous morphology, which benefits the high electron-transport performance. "The authors should add citations. I am also wondering if these descriptions are correct, generally, the crystalline of the films will lead the high high electron-transport performance.
- 7) What are the small peaks before the peak of 2a of the Figure 7a?
- 8) I would suggest to add the ref "J. Am. Chem. Soc. 2010, 132, 4, 1377–1382. "

Reviewer #3 (Remarks to the Author):

In this manuscript, Matsuo et al. show a highly efficient copper bromide promoted one-step direct oxidation of alkoxy to ketone with the aid of a fullerene pendant and systematically study the mechanisms via isotope-labelling experiments, kinetic studies and density functional theory (DFT) calculations of this direct oxidation reaction. Finally, a preliminary study, such as thermogravimetric

analyses (TGA), trap density, and electron mobility were carried out to estimate the performance of both the indano[60]fullerene 1a and the produced fullerene-fused ketone 2a as electron-transport layer through the vacuum-deposition process. This work is interesting and I recommend acceptance of this manuscript after minor revisions.

1. The authors demonstrated that this reaction can be obtained in equally high yield under both argon atmosphere and under ambient environment. Why is this reaction environmentally insensitive? When the reaction carried out in ambient environment under such a high temperature (100 °C), could C60 be oxidized? I suggest to add the high-resolution mass spectrum (HRMS) starting at 720 molecular weight to verify this.
2. In the experiments, the authors studied the effect of different substrate scope of the one-step oxidation reaction. Why did different substrate scopes no matter with electron-withdrawing substituent and electron-donating substituent have similar yields? As far as we know, substituent groups with different properties should have a great influence on the reactivity.
3. Energy level is an important parameter for novel fullerene derivatives, and it is the basic criteria to estimate whether this material could be applied as an electron transfer layer to facilitate the charge transfer. I strongly suggests the authors to measure the energy level (HOMO and LUMO energy levels) of these fullerene derivatives.
4. Authors mentioned that "the high-performance fullerene electron transport materials have never been achieved using vacuum-deposition process". However, C60 is a widely used electron transport materials by using vacuum-deposition process with high device performance.

Response to the comments from Reviewer 1:

In this manuscript, the authors reported the synthesis of fullerene-fused ketone via one-step reaction. The optimization of reaction conditions, scope of the reaction, and reaction mechanism are well studied and presented. This is a nice work and should be of interests to chemistry and organic electronics community. Therefore, I am happy to recommend the publication of this manuscript.

Minor revision suggestion: In Table 1, It is confused the readers of “t h” in the reaction condition.

→Thank you for the comment. The confusing “t h” abbreviation is revised to “time h” in Table 1.

Response to the comments from Reviewer 2:

In this manuscript, Yutaka Matsuo et. al developed a highly efficient copper bromide promoted one-step direct oxidation of alkoxy to with the aid of a fullerene pendant. They further investigate the mechanism of the reaction, their results indicating that fullerene can serve as an electron poor proceeding the one-step oxidation of alkoxy group to ketone. They also prepared the thin films of the fullerene-fused ketone, affording the pin-hole free and amorphous electron-transport layer and measured the electron-transport mobility by SCLC method. Overall, the synthesis method is simple while efficient, the characterization is comprehensive. I suggest to be published in communications chemistry after these comments have be addressed.

1) The comparison of the proton NMR of the **1a** and **2a** should be added.

→Thank you for the valuable advice. The discussion of the comparison is included in Mechanistic studies with additional figure in Supplementary Information. The revised content is attached below:

Moreover, the ¹H NMR peaks depicted at the typical aromatic region indicates the obvious down-field shift of **2a** compared to **1a**, which is ascribed from strong electron-deficiency of carbonyl group in **2a** (See Supplementary Fig. 6 for details).

2) What is the temperature of the NMR of the **2a**, **2b**, **2c** and **2d** have been done should be added in the SI, at room temperature or high temperature?

→Thank you for the suggestion. All the NMR measurements were carried out at room temperature. The temperature information for NMR measurement was added in the SI at "5. Experimental procedures". The revised content is attached below:

All NMR spectra were taken at room temperature by 400 MHz (Bruker AVANCE III 400 spectrometer), 500 MHz (Bruker AVANCE III 500 spectrometer) or 600 MHz (Bruker AVANCE III 600 spectrometer).

3) The TGA curve of **1a** is weird, can the authors explain what is the first step of the curve?

→Thank you for the comment. The first-step decomposition of **1a** at the 258.9 °C could be also explained by HPLC chart of thermally evaporated **1a** in Fig. 7 a). Although the thermal treatment on **1a** afforded

2a as a major component, a mixture of decomposed compounds obviously accounts for more than half based on the integration area in HPLC. Consequently, the TGA curve of **1a** with a 18.3% weight loss at first step could not give an accurate interpretation for the conversion of **1a** to **2a**. To provide a clear understanding, further discussion is added to the manuscript at “Performance of evaporable fullerene-fused ketone”. The additional content is attached below:

HPLC analysis of vacuum-deposited **1a**-film indicated that **1a** is instable in the vacuum-deposition process, which affords a mixture of thermally decomposed compounds (Fig. 7a). Meanwhile, thermogravimetric analyses (TGA) showed that **1a** have an initial decomposing temperature at 258.9 °C with a 18.3% weight loss (See Supplementary Fig. 9a for details). In well accordance with the HPLC analysis of vacuum-deposited **1a**-film, such dramatically high weight loss ratio of **1a** can be attributed to the complicate mixture of decomposed compounds.

4) I suggest to add the absorption spectra, the CV data and the energy levels of the fullerene, **1a** and **2a** both in solution and thin film, and make a comparison and explanation if there are different.

→Thank you for the comment. The CV data and the energy levels of substrates **1a-d** were directly cited from our previous research. For the CV data of products **2a-d** were measured using same experimental conditions in our previous research to ensure a reasonable comparison. We found fullerene-fused ketones **2a-d** have a lower LUMO energy level than **1a-d**, which is attributed to the electron-withdrawing property of directly bonded carbonyl group. In addition, the HOMO and LUMO energy levels of **2a-d** were also obtained using DFT calculations. Now, the additional discussion regarding the energy levels is included in the manuscript, and the related CV data, energy levels are added in the Supplementary Information with necessary literature citations. The details are attached below:

Besides the electron-transport mobility comparison of **2a**, the lowest unoccupied molecular orbital (LUMO) energy level of **2a** is also evaluated through cyclic voltammetry (See Supplementary Table 9 and 10 for details). Compared with LUMO levels of alkoxy substrates **1**, fullerene-fused ketones **2** show ~0.1 eV lower LUMO levels due to the strong electron-withdrawing property of carbonyl group. It should be noted that fullerene-fused ketones indicate the lowest LUMO energy levels among fullerene derivatives by far, which doubtlessly are compatible with common perovskite materials such as CH₃NH₃PbI₃. Meanwhile, DFT calculations demonstrate that fullerene-fused ketones have a deep enough highest occupied molecular orbital (HOMO) energy level, which is capable of blocking holes from recombining with electrons in photovoltaics (See Supplementary Table 11 for details).

Energy levels of evaporable fullerene-fused ketones 2a–d

Supplementary Table 9 Half-wave reduction potentials and LUMO levels of fullerenes.^a

Fullerenes	$E_{1/2}^{red}$ (V vs Fc/Fc ⁺)		LUMO level (eV) ^b
	E_1	E_2	
C ₆₀	-1.11	-1.50	-3.69
2a	-1.13	-1.51	-3.67
2b	-1.14	-1.51	-3.66
2c	-1.12	-1.50	-3.68
2d	-1.15	-1.52	-3.65

^aPotentials in eV vs a ferrocene/ferrocenium (Fc/Fc⁺) couple were recorded by cyclic voltammetry in *o*-DCB solution containing Bu₄N⁺(CF₃SO₂)₂N⁻ (0.1 M) as supporting electrolyte at 25 °C with a scan rate of 0.05 V/s. Platinum disk, platinum wire, and Ag/Ag⁺ electrodes were used as the working, counter, and reference electrodes, respectively. ^bEstimated LUMO levels using the following equation: LUMO level = -(4.8 + E_1) eV.⁶

Supplementary Table 10 Comparison of half-wave reduction potentials and LUMO levels for 1a–d and 2a–d.

1 ^a	E_1 (V)	LUMO level (eV)	2	E_1 (V)	LUMO level (eV)
1a	-1.20	-3.60	2a	-1.13	-3.67
1b	-1.21	-3.59	2b	-1.14	-3.66
1c	-1.19	-3.61	2c	-1.12	-3.68
1d	-1.23	-3.57	2d	-1.15	-3.65

^aHalf-wave potentials and LUMO levels of **1a–d** are cited from our previous research.⁷

Supplementary Table 11 Computed HOMO and LUMO levels of 2a–d.

Fullerenes	HOMO orbitals	HOMO levels (eV)	LUMO orbitals	LUMO levels (eV)
C ₆₀		-6.723		-3.893
PC ₆₁ BM		-6.319		-3.700
2a		-6.392		-3.805
2b		-6.356		-3.773
2c		-6.483		-3.885
2d		-6.316		-3.733

→ Thank you so much for the advice. Due to the poor solubility of fullerene-fused ketone in common organic solvent, we compared the UV-Vis spectra of C₆₀ and **2a** based on vacuum-deposition film, which more resembles real device application. As shown in UV-Vis spectra, **2a** indicated an obvious blue-shift at typical fullerene absorption peak 250 nm and ~330 nm, which can be attributed to the electron-withdrawing carbonyl group. Meanwhile, the absorption of **2a** at a range from 400 nm to 650 nm, where is a typical absorption range of perovskite materials, is much weaker than that of C₆₀, which indicates a potentially better light absorption for perovskite materials when using **2a** as electron-transporting layer in a normal-type structure. The additional discussion is attached below: In addition, the UV-vis spectra were applied to compare the visible light absorption of vacuum-deposition fabricated **2a**-film and C₆₀-film (See Supplementary Fig. 12 for details). The absorption spectrum of **2a**

showed an obvious blue-shift compared to that of C₆₀ at two typical fullerene absorption positions around 250 nm and 330 nm, which was further confirmed through DFT computation. This significant blue-shift is contributed by the strong electron-withdrawing carbonyl group in **2a**. More interestingly, compared to C₆₀, **2a** shows a much weak light absorption at a range of 400 nm ~ 650 nm, where is a typical absorption range for perovskite materials, providing a better light absorption for the perovskite materials when using **2a** as the electron-transporting layer in a normal-type structure.

Supplementary Fig. 12 UV-Vis spectra. **a)** C₆₀-film (black) and **2a**-film (red) through vacuum-deposition by 30 nm on bare glass substrate. **b)** DFT computational UV-vis of C₆₀ (black) and **2a** (red).

5) The authors also made the spin-coated and vacuum-deposited thin films, what solvent has been used to get the spin-coated thin films?

→Thank you for the valuable comment. As the fullerene-fused ketone has a poor solubility in common solvents, we prepared spin-coated **2a**-film by using saturated **2a** *ortho*-dichlorobenzene solution. The additional description is added in the SI and attached below:

TEM sample preparation. Spin-coated **2a**-film was prepared by spin-coating saturated **2a** *ortho*-dichlorobenzene with a 0.22 μm filter to remove any undissolved particles. After spin-coating onto the TEM grid, the grid was annealed at 100 °C to remove excess solvent. Vacuum-deposited **2a**-film was prepared directly from depositing **2a** by a thermal evaporator.

6) Between line 242 and 246, the authors mentioned that “Fig. 7c indicated that the spin-coated **2a**-film show obvious pinholes with substantial crystalline found in the selected area electron diffraction (See Supplementary Fig. 10a for details). In stark contrast, the vacuum-deposited **2a**-film exhibits a highly uniform and amorphous morphology, which benefits the high electron-transport performance. “The authors should add citations. I am also wondering if these descriptions are correct, generally, the crystalline of the films will lead the high electron-transport performance.

→Thank you for the comment. Generally, the perfect crystalline of the films, such as TiO₂, SnO₂ can give high electron-transport performance mainly because of few defects. However, considered fullerene-applied occasions, fullerene derivatives commonly have poor solubility and strong crystallinity, which easily results in pinholes when forming the film. This is because a combination of poor solubility and

strong crystallinity readily leads to cocrystals or even aggregates. Herein, as suggested by the reviewer, the related fullerene research is cited.

7) What are the small peaks before the peak of 2a of the Figure 7a?

→Thank you for the comment. Although there are nearly 4 notable peaks with shorter retention time than 2a, actual mixture shows more components than 4 species. Although it is difficult to get all characterizations for every decomposed compounds, at least we can assure that there is no remaining 1a (Retention time = ~ 7.5 min) and the peak after 2a is C₆₀. Accordingly, the note “C₆₀” are introduced to the Fig. 7a for a clear description and additional Supplementary Fig. 10 is included for comparing before/after deposition.

Supplementary Fig.10 HPLC analyses. a) Before deposition of 1a, b) Before deposition of 2a, c) After deposition of 1a, d) After deposition of 2a.

8) I would suggest to add the ref “J. Am. Chem. Soc. 2010, 132, 4, 1377–1382.”

→Thank you for the valuable advice. The sentence was revised with including the suggested literature as No.13 in references:

So far, fullerene derivatives have been demonstrated as versatile and high-performance n-type semiconductive materials in both organic solar cells and recently boosted perovskite solar cells,^{13–15} but the high-performance fullerene electron-transport derivatives have never been achieved using vacuum-deposition process.^{16–18}

Response to the comments from Reviewer 3:

In this manuscript, Matsuo et al. show a highly efficient copper bromide promoted one-step direct oxidation of alkoxy to ketone with the aid of a fullerene pendant and systematically study the mechanisms via isotope-labelling experiments, kinetic studies and density functional theory (DFT) calculations of this direct oxidation reaction. Finally, a preliminary study, such as thermogravimetric analyses (TGA), trap density, and electron mobility were carried out to estimate the performance of both the indano[60]fullerene **1a** and the produced fullerene-fused ketone **2a** as electron-transport layer through the vacuum-deposition process. This work is interesting, and I recommend acceptance of this manuscript after minor revisions.

1) The authors demonstrated that this reaction can be obtained in equally high yield under both argon atmosphere and under ambient environment. Why is this reaction environmentally insensitive? When the reaction carried out in ambient environment under such a high temperature (100 °C), could C₆₀ be oxidized? I suggest to add the high-resolution mass spectrum (HRMS) starting at 720 molecular weight to verify this.

→ Thank you for the comment. This environmental insensitivity of this reaction is mainly benefited from the strong air stability of fullerene cation species, which is also the state-of-the-art of this research. In this manuscript, besides including our previous studies reading fullerene cation species, we herein found fullerene-radical-cation mediated reaction demonstrate extremely high reactivity, which also contributes to the environmental insensitivity. Generally, C₆₀ is stable at a temperature below 300 °C under ambient environment. Consequently, 100 °C cannot reach the oxidation of C₆₀. However, considered the oxidants applied in this reaction, the referee reasonably suggested the extension of HRMS range to 720. Herein, we included full spectra of HRMS at a range of 700~1000 in the Supplementary Information, but no substantial peaks were found, which supports that no oxidation of fullerene occurred in this reaction.

2) In the experiments, the authors studied the effect of different substrate scope of the one-step oxidation reaction. Why did different substrate scopes no matter with electron-withdrawing substituent and electron-donating substituent have similar yields? As far as we know, substituent groups with different properties should have a great influence on the reactivity.

→ Thank you so much for the meaningful suggestion. Generally, electron negativity shows significant influence on reaction yields. According to the kinetics study, this one-step reaction shows a first-order reaction characteristic with a large rate constant as $7.8 \times 10^{-3} \text{ mol}^{-1} \text{ L}^{-1} \text{ s}^{-1}$ at 100 °C. In addition, the mechanistic study demonstrated that the rate determine step is the formation of the hemiketal structure

IV, which is mainly influenced by the reactivity at benzyl position. However, the benzyl position here is directly connected to fullerene cage and methoxy, which is far more reactive than conventional benzyl position. Accordingly, only one methoxy group or fluorine atom at benzene cannot significantly change the reactivity, which results in no substantial difference among different substrates. Moreover, the weak influence by substitutional group could be also found in the CV measurements. As the HOMO and LUMO are mainly located on the fullerene cage, the F- or MeO- attached on the benzene is too far to affect the fullerene cage.

3) Energy level is an important parameter for novel fullerene derivatives, and it is the basic criteria to estimate whether this material could be applied as an electron transfer layer to facilitate the charge transfer. I strongly suggests the authors to measure the energy level (HOMO and LUMO energy levels) of these fullerene derivatives.

→Thank you for the comment. The CV data and the energy levels of substrates **1a-d** were directly cited from our previous research. For the CV data of products **2a-d** were measured using same experimental conditions in our previous research to ensure a reasonable comparison. We found fullerene-fused ketones **2a-d** have a lower LUMO energy level than **1a-d**, which is attributed to the electron-withdrawing property of directly bonded carbonyl group. In addition, the HOMO and LUMO energy levels of **2a-d** were also obtained using DFT calculations. Now, the additional discussion regarding the energy levels is included in the manuscript, and the related CV data, energy levels are added in the Supplementary Information with necessary literature citations. The details are attached below:

Besides the electron-transport mobility comparison of **2a**, the lowest unoccupied molecular orbital (LUMO) energy level of **2a** is also evaluated through cyclic voltammetry (See Supplementary Table 9 and 10 for details). Compared with LUMO levels of alkoxy substrates **1**, fullerene-fused ketones **2** show ~0.1 eV lower LUMO levels due to the strong electron-withdrawing property of carbonyl group. It should be noted that fullerene-fused ketones indicate the lowest LUMO energy levels among fullerene derivatives by far, which doubtlessly are compatible with common perovskite materials such as $\text{CH}_3\text{NH}_3\text{PbI}_3$. Meanwhile, DFT calculations demonstrate that fullerene-fused ketones have a deep enough highest occupied molecular orbital (HOMO) energy level, which is capable of blocking holes from recombining with electrons in photovoltaics (See Supplementary Table 11 for details).

Energy levels of evaporable fullerene-fused ketones 2a–d

Supplementary Table 9 Half-wave reduction potentials and LUMO levels of fullerenes.^a

Fullerenes	$E_{1/2}^{red}$ (V vs Fc/Fc ⁺)		LUMO level (eV) ^b
	E_1	E_2	
C ₆₀	-1.11	-1.50	-3.69
2a	-1.13	-1.51	-3.67
2b	-1.14	-1.51	-3.66
2c	-1.12	-1.50	-3.68
2d	-1.15	-1.52	-3.65

^aPotentials in eV vs a ferrocene/ferrocenium (Fc/Fc⁺) couple were recorded by cyclic voltammetry in *o*-DCB solution containing Bu₄N⁺(CF₃SO₂)₂N⁻ (0.1 M) as supporting electrolyte at 25 °C with a scan rate of 0.05 V/s. Platinum disk, platinum wire, and Ag/Ag⁺ electrodes were used as the working, counter, and reference electrodes, respectively. ^bEstimated LUMO levels using the following equation: LUMO level = -(4.8 + E_1) eV.⁶

Supplementary Table 10 Comparison of half-wave reduction potentials and LUMO levels for 1a–d and 2a–d.

1 ^a	E_1 (V)	LUMO level (eV)	2	E_1 (V)	LUMO level (eV)
1a	-1.20	-3.60	2a	-1.13	-3.67
1b	-1.21	-3.59	2b	-1.14	-3.66
1c	-1.19	-3.61	2c	-1.12	-3.68
1d	-1.23	-3.57	2d	-1.15	-3.65

^aHalf-wave potentials and LUMO levels of 1a–d are cited from our previous research.⁷

Supplementary Table 11 Computed HOMO and LUMO levels of 2a–d.

Fullerenes	HOMO orbitals	HOMO levels (eV)	LUMO orbitals	LUMO levels (eV)
C ₆₀		-6.723		-3.893
PC ₆₁ BM		-6.319		-3.700
2a		-6.392		-3.805
2b		-6.356		-3.773
2c		-6.483		-3.885
2d		-6.316		-3.733

4) Authors mentioned that "the high-performance fullerene electron transport materials have never been achieved using vacuum-deposition process". However, C₆₀ is a widely used electron transport materials by using vacuum-deposition process with high device performance.

→Thank you for the valuable comment. The misleading description that mixing the [60]fullerene with fullerene derivatives was revised to have a precise definition of fullerene derivatives. Different to strong thermal stability of C₆₀, fullerene derivatives are prone to thermally decompose. The revised sentence is attached below:

but the high-performance fullerene derivatives electron-transport materials have never been achieved using vacuum-deposition process.^{16–18}

REVIEWERS' COMMENTS:

Reviewer #1 (Remarks to the Author):

The authors have fully addressed the concerns raised by this reviewer and the manuscript is acceptable.

Reviewer #2 (Remarks to the Author):

I am satisfied with the authors' revisions and recommended the publication of this manuscript in Communications chemistry.

Reviewer #3 (Remarks to the Author):

The authors have addressed my comments well, thus it can be accepted.